# Follicle-Stimulating Hormone Treatment and Male Idiopathic Infertility: Effects on Sperm Parameters and Oxidative Stress Indices according to *FSHR* c. 2039 A/G and c. -29 G/A Genotypes

**DOI:** 10.3390/jcm9061690

**Published:** 2020-06-02

**Authors:** Laura M. Mongioì, Rosita A. Condorelli, Angela Alamo, Rossella Cannarella, Nicolò Musso, Sandro La Vignera, Aldo E. Calogero

**Affiliations:** 1Department of Clinical and Experimental Medicine, University of Catania, Via S. Sofia 78, 95123 Catania, Italy; rosita.condorelli@unict.it (R.A.C.); angela.alamo1986@gmail.com (A.A.); rossella.cannarella@phd.unict.it (R.C.); sandrolavignera@unict.it (S.L.V.); acaloger@unict.it (A.E.C.); 2Bio-Nanotech Research and Innovation Tower (BRIT), University of Catania, Via S. Sofia 78, 95123 Catania, Italy; nmusso@unict.it

**Keywords:** follicle-stimulating hormone (FSH), male idiopathic infertility, conventional sperm parameters, bio-functional sperm parameters, oxidative stress, *FSHR* polymorphisms

## Abstract

Scientific evidence shows that the administration of follicle-stimulating hormone (FSH) to infertile patients with normal serum FSH concentrations improves sperm parameters in oligozoospermic men. The aim of this study was to evaluate the effects of highly purified urofollitropin (hpFSH) on conventional and bio-functional sperm parameters and on oxidative stress indices in patients with idiopathic infertility. We also evaluated the response to hpFSH on these parameters in relationship to *FSHR* c. 2039 A/G and *FSHR* c. -29 G/A genotypes. A prospective longitudinal study was conducted on 42 patients with idiopathic male infertility, 23 of whom underwent to *FSHR* c. 2039 A/G and *FSHR* c. -29 G/A genotyping. Each patient was asked to collect two semen samples before and after administration of 150 IU hpFSH three times a week for 16 weeks. Patients were divided into responders or non-responders based on whether their total sperm count had at least doubled or was less than double at the end of treatment, respectively. Responders showed a significantly higher semen volume, sperm concentration, spermatids, and leukocytes. Non-responders had a significant decrease of the percentage of spermatozoa in early apoptosis after hpFSH administration. Oxidative stress indexes did not differ significantly after FSH administration in both groups. Conventional and bio-functional sperm parameters did not differ in patients with *FSHR* c. 2039 GG and AA genotypes, and *FSHR* c. -29 GG genotype both before and after FSH administration. The *FSHR* c. 2039 and *FSHR* -29 G/A genotypes and allelic distribution did not differ between responders and non-responders. FSH showed to be capable of ameliorating sperm parameters in about half patients treated, therefore it may be helpful in patients with idiopathic infertility.

## 1. Introduction

The therapeutic effects of follicle-stimulating hormone (FSH) on spermatogenesis are well known. Scientific evidences show that the administration of FSH (both extractive purified FSH (pFSH) or recombinant FSH (rFSH)) to infertile patients with normal serum FSH concentrations, improve sperm parameters in oligozoospermic men and increase fertilization and pregnancy rates [1,2,3]. A position statement of the Italian Society of Andrology and Sexual Medicine (SIAMS) suggests that FSH administration increases sperm concentration in these patients [4]. On the contrary, some studies did not find significant effects of FSH on conventional sperm parameters, but showed an improvement of the degree of sperm DNA fragmentation [5,6,7], a decrease of the aneuploidy rate, and an increase in oocyte fertilization rate. However, data are still conflicting. Moreover, several single nucleotide polymorphisms (SNPs) have been currently identified in the *FSHR* and *FSHβ* genes. Some of them have been demonstrated to impact gene transcription (as it is the case of the *FSHβ* c. -211 G/T or the *FSHR* c. -29 G/A) or on signal transduction (e.g., the *FSHR* c. 2039 A/G), thus influencing male reproductive parameters such as FSH serum levels or testicular volume [8,9]. Also, it is known that the effectiveness of FSH therapy is modulated by polymorphisms of the *FSHR* [8] and *FSHβ* [9] genes. Accordingly, the percentage of responders to FSH therapy is significantly higher in *FSHβ* c. -211 TT male patients compared to GG carries. Finally, FSH is able to improve sperm DNA fragmentation depending on the *FSHR* c. 2039 G/A genotype, showing a greater efficacy in GG carries [8,9]. Hence, the analysis of these polymorphisms could be helpful in identifying the possibly responders to the treatment.

Some studies have reported evidence of a decrease in seminal reactive oxygen species (ROS) after FSH treatment [10], but more data on this topic are needed to draw a final conclusion.

The aim of this prospective longitudinal study was to evaluate the effects of highly purified urofollitropin (hpFSH) on conventional and bio-functional sperm parameters and on oxidative stress indexes in patients with idiopathic infertility. Moreover, the patients were genotyped to evaluate the *FSHR* c. 2039 A/G and *FSHR* c. -29 G/A single nucleotide polymorphism (SNP) distribution and their possible relationship with the response to treatment with FSH.

## 2. Patients and Methods

### 2.1. Patients

This was a preliminary study. We enrolled 42 patients with age ranging between 21 and 47 years (mean 34.9 ± 0.8 years) and idiopathic infertility lasting for more than 12 months (mean 29.5 ± 4.0 months).

At enrolment, all patients underwent medical history, physical examination, and hormonal assessment (FSH, luteinizing hormone –LH-, total testosterone). We included in the study male patients with idiopathic infertility. We excluded female causes of infertility and the mean age of the female partners was 31.9 ± 0.80 years. We also carefully excluded from the study patients under 18 years, with azoospermia, high FSH serum levels (>8 mIU/mL), primary testicular diseases (i.e., cryptorchidism and varicocele), male accessory glands infections, central hypogonadism, systemic diseases, chronic exposition to environmental and/or occupational toxicants, intake of spermiotoxic drugs, smoking, and alcohol or drugs abuse.

All patients were prescribed treatment with hpFSH at the dosage of 150 IU three times a week for four months (maximum administration time allowed by note 74 of Italian Drug Agency). Thirty-three patients completed the treatment without side effects. Nine subjects dropped out the study for personal reasons. Twenty-three patients underwent a blood sample to perform genotyping for the *FSHR* c. 2039 A/G SNP.

The protocol was approved by the Ethical Committee (EC) of the University Teaching Hospital “Policlinico-Vittorio Emanuele”, University of Catania (Protocol n.104/2016/PO of the opinions of the EC Registry), Catania, Italy. All procedures performed in studies involving human participants were in accordance with the ethical standards of the institutional and/or national research committee and with the 1964 Helsinki declaration and its later amendments or comparable ethical standards. An informed written consent was obtained from each patient.

### 2.2. Semen Analysis and Bio-Functional Sperm Parameter Evaluation

Each patient was asked to collect a semen sample before and after three months of hpFSH administration. Sperm analysis was conducted according to the World Health Organization (WHO) 2010 criteria [11]. The following bio-functional sperm parameters were evaluated by flow cytometry: percentage of alive and apoptotic spermatozoa, evaluation of mitochondrial membrane potential, degree of chromatin compactness/DNA fragmentation and seminal oxidative stress indexes as lipid peroxidation (LP), and the mitochondrial superoxide concentrations.

Flow cytometry analysis was performed using flow cytometer CytoFLEX (Beckman Coulter Life Science, Milan, Italy). CytoFLEX is equipped with two solid state laser at 488 nm and 638 nm and with seven fluorescence channels: 525/40 BP, 585/42 BP, 610/20 BP, 690/50 BP, 780/60 BP for excitation at 488 nm, and 660/10 BP, 712/25 BP, 780/60 BP for excitation at 638 nm. Data were analyzed by the software CytExpert1.2.

### 2.3. Evaluation of Sperm Apoptosis/Vitality

The externalization of phosphatidylserine (PS) on the outer cell surface is used as an indicator of early apoptosis. The assessment of PS externalization was performed using annexin V, a protein that binds selectively the PS in presence of calcium ions, fluorescein isothiocyanate (FITC)-labeled. Therefore, marking simultaneously the cells with annexin V and PI, we distinguished alive (with intact cytoplasmic membrane), apoptotic, or necrotic spermatozoa. Staining with annexin V and PI was obtained using a commercially available kit (Annexin V-FITC Apoptosis, Beckman Coulter, IL, Milan, Italy). An aliquot containing 0.5 × 10^6^/mL was suspended in 0.5 mL of buffer containing 10 µL of annexin V-FITC and 20 µL of PI and incubated for 10 min in the dark. After incubation, the sample was analyzed by the fluorescence channels 525/40 BP (FITC) and 585/42 BP, 610/20 BP, and 690/50 BP (PI). The different pattern of staining allowed to identify the different cell populations: FITC negative and PI negative indicate alive sperm cells, FITC positive and PI negative indicate spermatozoa in early apoptosis, and FITC positive and PI positive indicate sperm cells in late apoptosis.

### 2.4. Evaluation of the Mitochondrial Membrane Potential

The percentage of spermatozoa with low mitochondrial membrane potential (MMP) was evaluated using the lipophilic probe 5,5′,6,6′-tetrachloro-1,1′,3,3′tetraethyl-benzimidazolylcarbocyanine iodide (JC-1) able to selectively penetrate into mitochondria where it is in monomeric form, emitting at 527 nm. Therefore, JC-1 excited at 490 nm is able to form aggregates emitting at 590 nm in relation to the membrane potential. When the mitochondrial membrane becomes more polarized, the fluorescence changes reversibly from green to orange. In cells with normal membrane potential, JC-1 is in the mitochondrial membrane in form of aggregates emitting in an orange fluorescence, while in the cells with low membrane potential it remains in the cytoplasm in a monomeric form, emitting a green fluorescence. In regards the sample preparation, we incubated an aliquot containing 1×10^6^/mL spermatozoa with JC-1 (JC-1 Dye, Mitochondrial Membrane Potential Probe, DBA s.r.l, Milan, Italy) for 10 min, at a temperature of 37 °C and in the dark; after 10 min of incubation, the cells were washed in PBS and analyzed by the fluorescence channels 525/40 BP (FITC) and 585/42 BP (PE).

### 2.5. Assessment of DNA Fragmentation

The evaluation of DNA fragmentation was performed by the TUNEL assay (terminal deoxyuridine nick end labeling assay). This method uses Terminal deoxynucleotidyl Transferase (TdT), an enzyme that polymerizes at the level of DNA breaks, modifying nucleotides conjugated to a fluorochrome. The TUNEL assay was performed by using a commercially available kit (Apoptosis Mebstain kit, DBA s.r.l, Milan, Italy). To obtain a negative control, TdT was omitted from the reaction mixture; the positive control was obtained pre-treating spermatozoa (about 0.5 × 10^6^) with 1 mg/mL of deoxyribonuclease I, not containing RNAse, at 37 °C for 60 min prior to staining. The reading was performed by flow cytometry using the 525/40 BP fluorescence channels.

### 2.6. Degree of Chromatin Compactness Assessment

Chromatin compactness assessment was evaluated after a process of cell membrane permeabilization; in this way fluorophore was able to penetrate in the nucleus. An aliquot of 1 × 10^6^ spermatozoa was incubated with LPR DNA-Prep Reagent containing 0.1% potassium cyanate, 0.1% NaN3, non-ionic detergents, saline and stabilizers (Beckman Coulter, IL, Milan, Italy), in the dark at room temperature for 10 min and incubated with Stain DNA-Prep Reagent containing 50 µg/mL of propidium iodide (PI; <0.5%), RNase A (4 KUnitz/mL), <0.1% NaN3, saline, and stabilizers (Beckman Coulter, IL) in the dark at room temperature for 30 min. The samples were analyzed by cytometer using 585/42 BP and 610/20 BP fluorescence channels. The PI enters the cells, after adequate permeabilization of the cell membrane, and the more the chromatin is compact the less it can bind to it.

### 2.7. Sperm Membrane Lipoperoxidation Evaluation

LP evaluation was performed using the probe BODIPY (581/591) C11 (Invitrogen, Thermo Fisher Scientific, Eugene, OR, USA), which after being incorporated into cell membranes, responds to the attack of free oxygen radicals changing its emission spectrum from red to green. This change of the emission pattern is detected by the flow cytometer which provides an estimate of the degree of peroxidation. Briefly, 2 × 10^6^ spermatozoa were incubated with 5 mM of the probe for 30 min in a final volume of 1 mL. After washing with PBS, flow cytometry analysis was conducted using the 525/40 BP (FITC) and 585/42 BP (PE) fluorescence channels.

### 2.8. Measurement of Mitochondrial Superoxide Levels

Mitochondrial superoxide levels were detected by the MitoSOX red mitochondrial superoxide indicator (Invitrogen, Thermo Fisher Scientific, Eugene, OR, USA) that, once penetrated into the mitochondria, is rapidly oxidized by superoxide anion (not by the other free radicals), emitting a fluorescence which allows signal detection using the 525/40 BP (FITC) and 585/42 BP (PE) fluorescence channels by cytometer. Briefly, about 1 × 10^6^ spermatozoa were incubated with 5 mM of the probe for 30 min in a final volume of 1 mL. After washing with PBS, flow cytometry analysis was conducted using the 525/40 BP (FITC) and 585/42 BP (PE) fluorescence channels.

### 2.9. DNA Extraction

Genomic DNA was extracted from blood cells using the PureLink^®^ Genomic DNA Kits (invitrogen Catalog Numbers K1821-04) for purification of genomic DNA according to the manufacturer’s instructions. The concentration and the quality of the DNA was determined using a ND-1000 spectrophotometer (NanoDrop, Thermo Scientific, USA). Allelic Discrimination was performed with TaqMan assay in order to show a different genotyping distribution of *FSHR* polymorphisms. Probes and primers for the *FSHR* c. 2039 A/G (rs6166) and *FSHR* c. -29 G/A (rs1394205) polymorphisms were chosen on https://www.thermofisher.com/it/en/home/life-science/pcr/real-time-pcr/real-time-pcr-assays/snp-genotyping-taqman-assays.html?SID=fr-taqman-2. The reaction was carried out according to manufacturer’s instructions (cod 4371355, Applied Biosystems, CA, USA). Each DNA sample was analyzed in triplicate [12]. Allelic Discrimination real-time PCR analysis was performed using LightCycler^®^ 480 System (Roche Molecular Systems, Inc, Pleasanton, CA, USA).

### 2.10. Statistical Analysis

The results are reported as mean ± SEM (standard error of the mean) throughout the study. Statistical analysis of the data was performed using Student’s *t*-test in the entire cohort. Subsequently, patients were divided into two groups on the basis of their total sperm count response to FSH administration. As previously reported [13], patients with total sperm count equal or greater than twice their pre-treatment values were considered responders to treatment, whereas those who had this parameter lower than twice were considered non-responders. The data resulting from this classification were analyzed by one-way analysis of variance (ANOVA) followed by the Duncan’s Multiple Range test. SPSS 22.0 for Windows was used for statistical analysis (SPSS Inc., Chicago, IL, USA). The results with a *p*-value less than 0.05 were considered statistically significant.

## 3. Results

We did not find any statistical differences in conventional sperm parameters after treatment in all patients (Table 1). After subdivision into two groups, the group of responders (*n* = 15, 45.5%) showed an increased semen fluid volume after hpFSH treatment (*p* = 0.02; Figure 1). In addition, these patients had higher sperm concentration compared to baseline (*p* = 0.001; Figure 1) and eight patients (53.3%) achieved values within the normal range. Moreover, these patients, after therapy with hpFSH, had a significantly increase of the percentage of spermatids (*p* = 0.04) and leukocyte concentration (*p* = 0.04; Figure 1). The other parameters did not differ significantly compared to pre-treatment values (Figure 1). In the group of non-responders (*n* = 18, 55.5%), sperm conventional parameters did not change significantly after FSH administration (Table 2). Four patients (22.2%) showed sperm concentration and total sperm count within the normal range after hpFSH administration.

Bio-functional sperm parameters and oxidative stress indicators were evaluated only in 22 patients, because a too low sperm number did not allow to measure these parameters in the remaining patients. The responder group (36.4%) did not show any significant change compared with pre-treatment values in the percentage of alive, early and/or late apoptotic spermatozoa and in the percentage of spermatozoa with low MMP and decreased chromatin compactness, although the percentage of spermatozoa with fragmented DNA decreased by about 53.9% at the end of treatment (Table 3). In the non-responder group (63.6%), we found a statistically significant decrease of the percentage of early apoptotic cells (*p* = 0.04; Figure 2). The other bio-functional sperm parameters of non-responders are reported in Figure 2.

As for the oxidative stress markers, we did not find significant differences of LP and mitochondrial superoxide anion concentrations between pre- vs. post-treatment values in the responder group (Figure 3). In this group of patients, mitochondrial superoxide anion increased by about 62.3% at the end of the treatment (Figure 4). In non-responders, LP and mitochondrial superoxide anion decreased respectively by about 54.7% and 10.1% vs. pre-treatment, but the difference did not reach the statistical significance (Figure 3 and Figure 4).

Lastly, as secondary outcome, at the end of the observation period four couples (12.1%) got pregnant, two spontaneously and two after intrauterine insemination (IUI). Only one male partner of these couple was part of the responders group.

Table 4 and Table 5 summarize our findings on *FSHR* c. 2039 A/G and *FSHR* c. -29 G/A SNPs. Conventional and bio-functional sperm parameters did not differ in GG vs. AA patients both at baseline and after FSH administration. The *FSHR* c. 2039 and *FSHR* c. -29 genotypes and allelic distributions did not differ among responders and non-responders. Distribution of *FSHR* c. 2039 A/G genotype did not differ among responders and not-responders. The frequency of the *FSHR* c. -29 GG, genotype, as well as that of the G allele, were significantly higher compared to that of the AA genotype and A allele in both responder and non-responder groups, thus reflecting the higher prevalence of the GG genotype and of the G allele, compared to the AA genotype and the A allele in the patients enrolled in this study.

## 4. Discussion

### 4.1. Conventional Sperm Parameters

The role of gonadotropins in the treatment of male idiopathic infertility has been largely explored, but conclusive data are still lacking. A systematic review [14] and a meta-analysis [3] reported an improvement in sperm parameters and a higher pregnancy rate (both spontaneous or subsequently assisted reproduction technology) after treatment with FSH in oligozoospermic patients. Nevertheless, although FSH treatment could be useful for its role in stimulating spermatogenesis and mitotic and meiotic DNA synthesis, there is no conclusive data about beneficial efficacy for the treatment of male idiopathic infertility. This is likely due to several factors, such as the variability of enrolment criteria, the different interpretation of seminal parameters, and the different dosages and treatment lengths used.

An initial analysis of sperm conventional parameters of all enrolled patients did not show a significant improvement after treatment with hpFSH. However, a closer look at the results from each patient suggest that some patients seemed to clearly respond to the treatment, whereas others did not. Hence, we evaluated the results by classifying the patients into two groups, responders and non-responders, according to the doubling of their total sperm count, as previously published [13]. It is not known why some patients respond to FSH treatment and others do not, but certainly the effectiveness of therapy is influenced by polymorphisms of the *FSHR* [8] and *FSHβ* [9] genes.

In our study we found that in the group of responders, sperm concentration significantly improved after three months of hpFSH administration. The same group showed a statistically significant increase in seminal fluid volume and spermatid count; the latter is an index of tubular function and stimulation of the spermatogenic process. A previous study reported that spermatid count in the seminal fluid is related to testicular cytology, thus it could be a predictive parameter of response to FSH administration [15]. It is important to note that we also found a significant increase of seminal fluid leukocyte concentration in responder patients. This evidence is in agreement with our previously observation that CD45-positive cells (pan-leukocyte marker) showed a positive linear correlation with the percentage of spermatids [16].

The effectiveness of FSH administration in patients with idiopathic infertility appears to be dose-dependent [4]. In 2002, Foresta and colleagues showed that the treatment with recombinant human FSH (rhFSH) at the dosage of 50 IU on alternate days did not have any effect on conventional sperm parameters [17]. On the contrary, the administration of rhFSH 100 IU induced a significant improvement of sperm concentration [17]. These results are supported by several studies in the literature which state that hpFSH is effective in improving sperm concentration when given at a weekly cumulative dose <450 IU, e.g., 75 IU every other day or 150 IU three times a week [7,10,18,19]. Very recently, we published a meta-analysis of randomized controlled studies to evaluate FSH dosage-effect on conventional sperm parameters. The results of this study showed that at high doses (700–1050 IU/week), FSH administration increased sperm concentration, total sperm count, and progressive motility, while normal forms showed only a trend to an increase [20].

### 4.2. Bio-Functional Sperm Parameters

Many authors have not found a significant increase in sperm parameters after FSH administration [1,2,21,22]. Nevertheless, there is evidence to support FSH treatment in idiopathic infertile patients with oligozoospermia and serum FSH concentration within the normal range, as a valid strategy to improve sperm quality [4].

Therefore, even if substantial variations of conventional sperm parameters were not observed after hpFSH therapy, it is possible to obtain an improvement in sperm quality. In fact, the non-responders to FSH showed a significant decrease of the percentage of early apoptotic spermatozoa, although the other parameters did not significantly differ from pre-treatment values. The degree of DNA fragmentation was more than halved after hpFSH treatment, but the data did not reach statistical significance.

No study has so far evaluated the effects of FSH on sperm MMP and apoptosis. Conversely, some studies have assessed DNA fragmentation in idiopathic oligozoospermic patients treated with FSH. In 2010, a prospective randomized controlled study evaluated DNA fragmentation index (DFI) before and after three months of rFSH treatment [6]. The results of this study showed that DFI decreased significantly after rFSH treatment. Ruvolo and colleagues also evaluated DFI in 53 men with hypogonadotropic hypogonadism and idiopathic oligoasthenoteratozoospermia before and after 90 days of r-FSH administration. The authors concluded that DFI was significantly reduced after therapy in patients with basal DFI > 15% [23]. However, these data, opposed to our results, could be explained by the different methodology used for evaluation of DNA fragmentation and by the type of FSH administered. Moreover, a multicenter longitudinal prospective open-label trial showed that the significant reduction in DFI was evident in some carriers of a specific polymorphism of *FSHR* [24]. Another study showed a significant reduction of DNA fragmentation after 90 days of hpFSH administration [10]. More recently, Garolla and collaborators, after FSH therapy, found a significant improvement of conventional sperm parameters and DNA fragmentation, the last one assessed through the TUNEL assay [7]. Nevertheless, the diversity of the results obtained can be attributed to the lack of unique cut-offs for the TUNEL test [25].

### 4.3. Oxidative Stress Indexes

To our knowledge, this is the first study that has evaluated the effects of FSH administration on sperm LP and mitochondrial superoxide concentrations, indexes of oxidative stress. The assessment of ROS as oxidative stress markers in idiopathic male infertility could be an important source of information and their evaluation is helpful to the clinician to decide the best therapeutic option for each individual case. Moreover, ROS are physiologically produced by spermatozoa and they are useful for intra and extracellular communications [26]. The present study showed that the percentage of spermatozoa with membrane LP did not change after FSH administration in both groups analyzed. The use of BODIPY allow us to obtain a direct quantification of LP, whereas in the past only indirect methods were used to measure the final products of the lipoperoxidation process, such as malondialdehyde [27,28]. As for mitochondrial superoxide concentrations, they increased by about 62.3% in responders to FSH administration. This result might be related to the increased number of leukocytes and spermatids, both sources of superoxide anion.

Superoxide anion is the principal ROS generator in cells and it plays a crucial role in sperm capacitation and acrosome reaction. To confirm this, in 1995 de Lamirande and Gagnon showed that superoxide anion exogenously administered induced sperm capacitation and capacitating spermatozoa produced themselves high concentrations of this anion. Moreover, removing ROS by administering enzyme superoxide dismutase (SOD), sperm capacitation is prevented [29]. A similar result was obtained by another study, in which the addition of SOD before the capacitation or 15 before the induction of acrosome reaction, reduced the acrosome reaction itself [30]. On this account, the authors concluded that superoxide anion was necessary for sperm capacitation, but alone it is not enough for it to happen. Therefore, we can hypothesize that the increased expression of mitochondrial superoxide anion may not necessarily be an indication of cellular damage from oxidative stress, but, rather, it could be a sign of therapeutic success, since reactive oxygen species (ROS) are necessary for the maturation and capacity of spermatozoa.

In addition, as a secondary outcome, at the end of the three months of treatment we registered two pregnancies following intrauterine insemination (IUI) and two spontaneous pregnancies. Many studies have reported an improved pregnancy rate after FSH administration [3] and FSH administration is suggested in couples with idiopathic male infertility factor to ameliorate spontaneous and following assisted reproductive technique pregnancy rate [4].

### 4.4. FSHR Polymorphisms

The *FSHR* maps in the 2p16.3 chromosome and contains 10 exons and 9 introns. The extracellular domain of the FSHR is encoded by the first nine exons, while the C-terminal region of the extracellular domain, the trans-membrane and the intracellular domains are encoded by the exon 10 [31]. More than 900 *FSHR* SNPs are actually listed in the HapMap database (http://hapmap.ncbi.nlm.nih.gov). A large body of evidence has been released so far on the c. 2039 A/G (p. Asp680Ser; rs 6166) *FSHR* SNPs, mapping in the exon 10, into the transmembrane domain. This SNP is known to influence the efficiency of signal transduction. Particularly, the c. 2039 A/G genotype impacts on the expression of the amino acid 680, being the FSHR Ser680Ser (GG) more resistant to the FSH signal compared with the Asp680Asp (AA) [32]. Patients carries of the 680Ser show lower testicular volume (a decrease of 1.40 mL), total testosterone, inhibin B and increased FSH serum levels compared to Asp680 carries [33]. Only one study, carried out in 105 infertile patients, has assessed the role of the *FSHR* c. 2039 A/G on responsiveness to FSH therapy by evaluating the conventional sperm parameters. The aplotype Ala307-Ser680 ensured the best responsiveness in terms of total sperm count, sperm progressive motility and morphology [34]. In contrast, our finding did not confirm these results and further data are needed to evaluate whether a correlation between these polymorphisms and the response to FSH administration exists. The *FSHR* -29 G/A SNP has been reported to influence *FSHR* transcription. In particular, the A allele is associated with a 56% lower *FSHR* transcriptional activity in-vitro [35]. Only one study has assessed the effect of this SNP on male reproductive function so far [8], and its possible predictive role to FSH responsiveness has never been assessed. To our knowledge, this is the first study describing FSH responsiveness in patients genotyped for this SNP. We found a significantly higher distribution of the GG genotype and the G allele, compared to the AA genotype and the A allele, and no predictive role of *FSHR* -29 G/A SNP on FSH responsiveness.

## 5. Conclusions

The use of FSH for the treatment of male idiopathic infertility is still debated and several reasons could explain the different results obtained. Our results showed that hpFSH could be a valid therapy for oligozoospermic patients, resulting in an improvement of both conventional and bio-functional sperm parameters in about half to the patients receiving such a treatment. Moreover, even in those patients we called non-responders, a slight improvement in the semen quality could be achieved.

Thus, we suggest that FSH is a valid therapeutic alternative in patients with idiopathic oligozoospermia and normal FSH values, but other studies are needed to further explore its effects and the relationship with *FSHR* gene SNPs.

## 6. Strengths and Limitations

This is the first study to evaluate the effects of FSH administration on seminal markers of oxidative stress in patients genotyped for *FSHR* c. 2039 A/G and *FSHR* c. -29 G/A SNP. It is also the first study describing FSH responsiveness in patients genotyped for the *FSHR* c. -29 G/A polymorphism. These are undoubtedly elements of novelty.

The limitation of the study is the small sample size, however, in our opinion, this aspect is justified by very strict exclusion criteria, as reported in the section of the methods and by the laboratory protocol that we have used, which evidently is not addressed to all infertile patients according to the current international guidelines [36]. Moreover this being a clinical study preliminary provides limited costs.

## Figures and Tables

**Figure 1 jcm-09-01690-f001:**
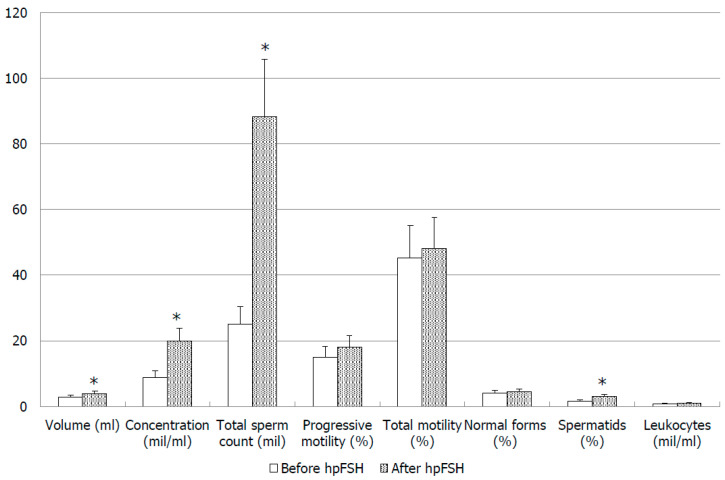
Conventional sperm parameters before and after treatment with high purified follicle-stimulating hormone (hpFSH) in infertile responder patients, * *p* < 0.05.

**Figure 2 jcm-09-01690-f002:**
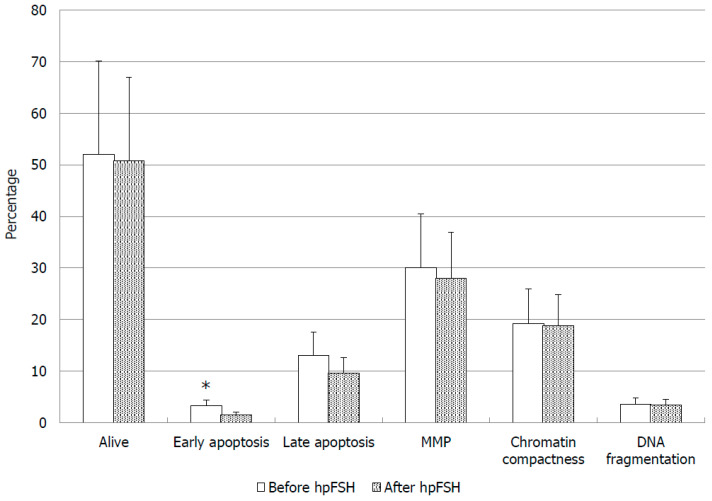
Bio-functional sperm parameters before and after treatment with high purified FSH (hpFSH) in infertile non-responder patients. * *p* < 0.05.

**Figure 3 jcm-09-01690-f003:**
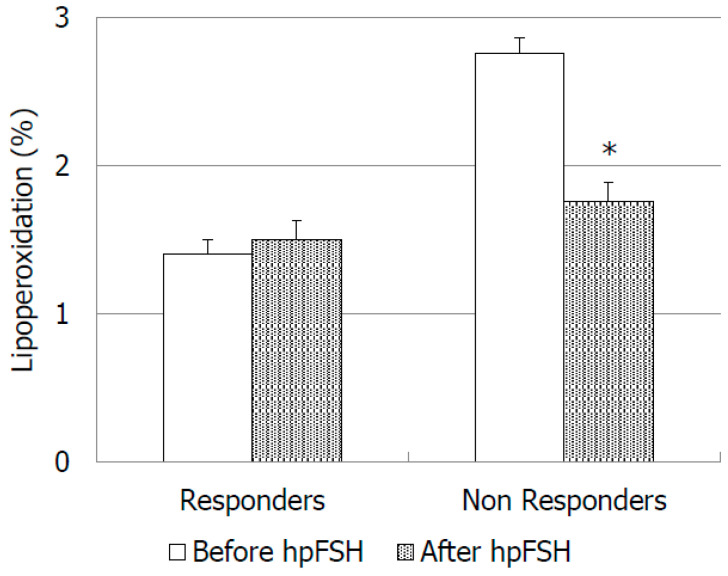
Percentage of spermatozoa with lipoperoxidation before and after treatment with high purified FSH (hpFSH) in infertile responder and non-responder patients. * *p* < 0.05.

**Figure 4 jcm-09-01690-f004:**
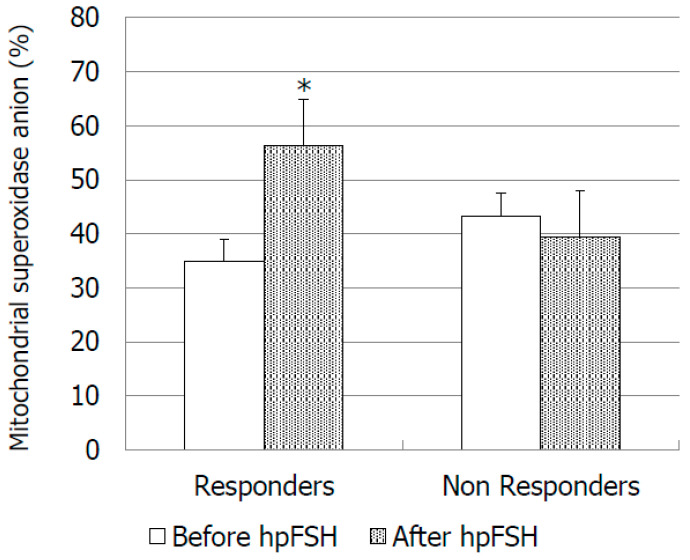
Percentage of spermatozoa with mitochondrial superoxide anion expression before and after treatment with high purified FSH (hpFSH) in infertile responder and non-responder patients. * *p* < 0.05.

**Table 1 jcm-09-01690-t001:** Conventional sperm parameters before and after treatment with follicle-stimulating hormone (hpFSH) in the entire patient cohort.

Parameter	Before hpFSH	After hpFSH	*p*-Value
Volume (mL)	2.77 ± 0.21	2.83 ± 0.27	>0.05
Concentration (mil/mL)	12.20 ± 2.13	14.60 ± 0.46	>0.05
Total sperm count (min/ejaculate)	30.87 ± 5.97	53.26 ± 13.72	>0.05
Progressive motility (%)	14.47 ± 2.03	14.51 ± 2.28	>0.05
Total motility (%)	47.52 ± 2.60	45.69 ± 3.17	>0.05
Normal forms (%)	4.14 ± 0.43	4.03 ± 0.41	>0.05
Spermatids (%)	1.67 ± 0.20	3.66 ± 1.20	>0.05
Leukocytes (mil/mL)	0.49 ± 0.24	0.63 ± 0.19	>0.05

**Table 2 jcm-09-01690-t002:** Conventional sperm parameters before and after hpFSH treatment in non-responder patients.

Parameter	Before hpFSH	After hpFSH	*p*-Value
Volume (mL)	2.56 ± 0.28	2.64 ± 0.36	>0.05
Concentration (mil/mL)	14.05 ± 3.11	10.17 ± 2.26	>0.05
Total sperm count (mil/ejaculate)	31.66 ± 6.87	24.15 ± 4.61	>0.05
Progressive motility (%)	10.94 ± 1.63	11.61 ± 1.54	>0.05
Total motility (%)	43.8 ± 3.379	43.72 ± 4.04	>0.05
Normal forms (%)	3.72 ± 0.52	3.72 ± 0.41	>0.05
Spermatids (%)	1.79 ± 0.24	4.22 ± 2.13	>0.05
Leukocytes (mil/mL)	0.28 ± 0.06	0.29 ± 0.97	>0.05

**Table 3 jcm-09-01690-t003:** Bio-functional sperm parameters before and after hpFSH treatment in responder patients.

Parameter	Before hpFSH	After hpFSH	*p*-Value
Alive spermatozoa (%)	53.57 ± 16.49	53.25 ± 10.93	>0.05
Spermatozoa in early apoptosis (%)	2.87 ± 1.33	2.8 ± 2.05	>0.05
Spermatozoa in late apoptosis (%)	17.55 ± 20.79	13.67 ± 8.03	>0.05
Spermatozoa with low MMP (%)	51.02 ± 16.874	46.14 ± 25.37	>0.05
Sperm chromatin compactness (%)	27.41 ± 10.92	25.73 ± 5.83	>0.05
Sperm DNA fragmentation (%)	10.56 ± 4.85	4.87 ± 3.01	>0.05

**Abbreviation**: MMP = Mitochondrial Membrane Potential.

**Table 4 jcm-09-01690-t004:** Conventional and bio-functional sperm parameters before and after hpFSH treatment in infertile patients divided according to the *FSHR* 2039A/G and *FSHR* G/A genotype.

	*FSHR* 2039A/G				*FSHR* -29 G/A		
	AA		GG		GG		AA
Parameter	Before hpFSH	After hpFSH	Before hpFSH	After hpFSH	Before hpFSH	After hpFSH	Before hpFSH
Volume (mL)	2.8 ± 0.5	3.0 ± 0.5	2.9 ± 0.4	3.4 ± 0.4	2.6 ± 1.1	2.8 ± 1.0	1.9 ± 0.5
Concentration (mil/mL)	12.3 ± 3.4	11.0 ± 3.7	12.2 ± 6.2	20.3 ± 11.5	16.7 ± 15.3	17.0 ± 17.7	4.8 ± 4.6
Total sperm count (min/ejaculate)	33.6 ± 9.1	36.0 ± 14.1	31.0 ± 15.5	85.6 ± 60.2	36.2 ± 32.1	53.2 ± 84.7	7.7 ± 6.2
Progressive motility (%)	18.0 ± 3.7	12.0 ± 2.9	17.7 ± 3.9	16.8 ± 5.3	15.7 ± 8.6	15.2 ± 9.8	20.0 ± 14.1
Total motility (%)	49.0 ± 6.2	46.0 ± 7.6	50.8 ± 4.1	55.2 ± 4.2	50.3 ± 13.3	48.2 ± 15.7	52.5 ± 10.6
Normal forms (%)	4.3 ± 0.6	4.5 ± 1.4	6.3 ± 1.6	4.4 ± 1.2	3.9 ± 2.2	4.5 ± 2.5	7.0 ± 7.1
Spermatids (%)	1.6 ± 0.5	3.0 ± 1.6	2.3 ± 0.4	3.6 ± 1.8	2.0 ± 1.6	2.9 ± 3.0	1.0 ± 0.0
Leukocytes (mil/mL)	0.4 ± 0.1	0.5 ± 0.2	0.2 ± 0.1	0.3 ± 0.1	0.4 ± 0.3	0.6 ± 0.5	0.2 ± 0.1
Alive spermatozoa (%)	57.5 ± 6.9	47.4 ± 4.6	51.6 ± 9.2	44.6 ± 5.2	51.2 ± 18.9	47.9 ± 12.8	---
Spermatozoa in early apoptosis (%)	4.1 ± 2.6	1.7 ± 0.6	2.3 ± 0.6	2.3 ± 1.2	2.0 ± 1.2	2.7 ± 2.5	---
Spermatozoa in late apoptosis (%)	11.0 ± 5.1	12.0 ± 1.6	12.3 ± 6.1	7.5 ± 2.9	18.3 ± 21.0	12.9 ± 10.4	---
Spermatozoa with low MMP (%)	26.3 ± 10.9	10.3 ± 0.9	43.9 ± 13.6	50.6 ± 14.3	30.7 ± 22.0	31.3 ± 24.6	---
Sperm chromatin compactness (%)	18.8 ± 9.9	23.9 ± 0.7	18.6 ± 2.0	15.2 ± 1.3	20.8 ± 13.1	19.2 ± 5.0	---
Sperm DNA fragmentation (%)	1.7 ± 0.5	4.7 ± 2.9	3.8 ± 0.6	4.2 ± 1.3	3.3 ± 2.6	3.3 ± 3.3	---

No *FSHR* -29 AA patient was assessed for bio-functional parameters. Conventional sperm parameters after hpFSH were unavailable in *FSHR* -29 AA patients.

**Table 5 jcm-09-01690-t005:** Distribution of the *FSHR* 2039A/G and *FSHR* G/A genotype in infertile responder and non-responder patients to FSH administration.

	*FSHR* 2039AG Genotype			A/G Allele Frequency		*FSHR* -29 G/A Genotype			G/A Allele Frequency	
	AA	AG	GG	A	G	AA	AG	GG	G	A
**Responders**	2/6 33.3%	2/6 33.3%	2/6 33.3%	6/12 50%	6/12 50%	---	0/4 0%	4/4 100.0 *%	8/8 100%	0/8 0.0 *%
**Non-responders**	3/11 27.3%	2/11 18.2%	6/11 54.5%	12/22 54.5%	10/22 45.5%	---	4/13 30.8%	9/13 69.2 *%	22/26 84.6%	4/26 15.4 *%

* *p* < 0.05 vs. AG (chi-squared test). No *FSHR* -29 AA patient was assessed for bio-functional parameters. Conventional sperm parameters after hpFSH were unavailable in *FSHR* -29 AA patients.

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
