# Peer review of "Follicle-Stimulating Hormone Treatment and Male Idiopathic Infertility: Effects on Sperm Parameters and Oxidative Stress Indices according to FSHR c. 2039 A/G and c. -29 G/A Genotypes"

_jcm, 2020, doi:10.3390/jcm9061690_

Round 1

Reviewer 1 Report

The paper is ready for the publication.

Author Response

Thank you very much for your comment

Reviewer 2 Report

The small sample size is much more than a minor limitation of the study as the authors suggest. It precludes any conclusion on the association of the genotypes with the effects of FSH.

Author Response

We are sorry, but we believe that the manuscript contains very original data. This is a preliminary study. We added this declaration in the Methods section.

Reviewer 3 Report

Ok there was a problem with the table: corrected

Basically there is no real modifications tto do

Author Response

Thank you for your comment.

This manuscript is a resubmission of an earlier submission. The following is a list of the peer review reports and author responses from that submission.

Round 1

Reviewer 1 Report

The FSH treatment in male infertility and especially in idiopathic cases is daubt and results are controversial. Despite the low number of patients treated in this paper, authors showed intersting reaults on bio-funcional sperm parameters and seminal markers of ossidative stress. it could suggest important infication for clinicians and suggest new research on this topic.

Reviewer 2 Report

The paper FSH treatment and male idiopathic infertility: effects on sperm parameters and oxidative stress indices according to FSHR c.2039A>G and c.-29G/A genotypes aims to evaluate the effect of known FSHr polymorphisms in sperm parameters and biomarkers in male idiopathic infertility.

Although this is a relevant idea it must be tested with higher number of patients as polymorphisms may have a role in susceptibility to FSH but the little number of patients tested (23) prevents any valid conclusion. 

After the classification of such a small number of patients in 2 groups (responders and non-responders) according to the genotypes the authors couldn't even find a man homozygous for the AA genotype of the FSHR-29G>A!

So, even though there are some ideas that deserve further exploitation such as the fact that increased expression of mitochondrial superoxide anion may be an indicator of therapeutic success rather than of cellular damage i suggest that the authors increase the sample at least 10 times.

Reviewer 3 Report

First of all the authors say” Scientific evidence……” , for LH and testosterone it is probably true but for FSH..

The least we can say it is rather uncertain:  for DFI decrease: Colacurci et al. :paper (abstract)     In the subgroup of patients with high basal DFI values (>15%), rFSH treatment significantly increased DFI (P < .01),This is not exactly what we can expect 

Santi study (abstract)…: limit the strength of these results.

This is obvious when we see the data in the tables;: a decrease in concentration and sperm count sperm count. Moreover when we look at the data in the overall and in the non responders: table 1 and 2

Overall concentration :           14.05 and then 10.0

Total count   31.66 and then 24.25

Non responders   concentration  14.05  and then 10.1 

Total count 31.66 and then 24.15

How is this possible if we have 15 responders+ and 18 non responders

E have a 45/55 ratio….it seems impossible to have the same values in the table

It looks for table 1 and 2 like a cut and paste

This opens the question, on which criteria the Responders++ and the responders—were selected. Especially when we look line 186-187 “ in the non responder group  (63.6 % ): this means 21 patients...

Hen several other points. For decondensation. The technique is not a standard, well recognize technique for sperm…Malondialdehyde is a good marker of lipid peroxidation, DNA fragmentation and decondensation.

The real interesting part is the polymorphism

In conclusion, this paper is not really informative, rather confusing and troubling. Totally impossible t understand table 1 and 2. The selection for Non responders and responders seems “moving” according  to the parameter tested

. Cannot be accepted without profound modifications

First of all the authors say” Scientific evidence……” , for LH and testosterone it is probably true but for FSH..

The least we can say it is rather uncertain:  for DFI decrease: Colacurci et al. :paper (abstract)     In the subgroup of patients with high basal DFI values (>15%), rFSH treatment significantly increased DFI (P < .01),This is not exactly what we can expect 

Santi study (abstract)…: limit the strength of these results.

This is obvious when we see the data in the tables;: a decrease in concentration and sperm count sperm count. Moreover when we look at the data in the overall and in the non responders: table 1 and 2

Overall concentration :           14.05 and then 10.0

Total count   31.66 and then 24.25

Non responders   concentration  14.05  and then 10.1 

Total count 31.66 and then 24.15

How is this possible if we have 15 responders+ and 18 non responders

E have a 45/55 ratio….it seems impossible to have the same values in the table

It looks for table 1 and 2 like a cut and paste

This opens the question, on which criteria the Responders++ and the responders—were selected. Especially when we look line 186-187 “ in the non responder group  (63.6 % ): this means 21 patients...

Hen several other points. For decondensation. The technique is not a standard, well recognize technique for sperm…Malondialdehyde is a good marker of lipid peroxidation, DNA fragmentation and decondensation.

The real interesting part is the polymorphism

In conclusion, this paper is not really informative, rather confusing and troubling. Totally impossible t understand table 1 and 2. The selection for Non responders and responders seems “moving” according  to the parameter tested

. Cannot be accepted without profound modifications

First of all the authors say” Scientific evidence……” , for LH and testosterone it is probably true but for FSH..

The least we can say it is rather uncertain:  for DFI decrease: Colacurci et al. :paper (abstract)     In the subgroup of patients with high basal DFI values (>15%), rFSH treatment significantly increased DFI (P < .01),This is not exactly what we can expect 

Santi study (abstract)…: limit the strength of these results.

This is obvious when we see the data in the tables;: a decrease in concentration and sperm count sperm count. Moreover when we look at the data in the overall and in the non responders: table 1 and 2

Overall concentration :           14.05 and then 10.0

Total count   31.66 and then 24.25

Non responders   concentration  14.05  and then 10.1 

Total count 31.66 and then 24.15

How is this possible if we have 15 responders+ and 18 non responders

E have a 45/55 ratio….it seems impossible to have the same values in the table

It looks for table 1 and 2 like a cut and paste

This opens the question, on which criteria the Responders++ and the responders—were selected. Especially when we look line 186-187 “ in the non responder group  (63.6 % ): this means 21 patients...

Hen several other points. For decondensation. The technique is not a standard, well recognize technique for sperm…Malondialdehyde is a good marker of lipid peroxidation, DNA fragmentation and decondensation.

The real interesting part is the polymorphism

In conclusion, this paper is not really informative, rather confusing and troubling. Totally impossible t understand table 1 and 2. The selection for Non responders and responders seems “moving” according  to the parameter tested

. Cannot be accepted without profound modifications

First of all the authors say” Scientific evidence……” , for LH and testosterone it is probably true but for FSH..

The least we can say it is rather uncertain:  for DFI decrease: Colacurci et al. :paper (abstract)     In the subgroup of patients with high basal DFI values (>15%), rFSH treatment significantly increased DFI (P < .01),This is not exactly what we can expect 

Santi study (abstract)…: limit the strength of these results.

This is obvious when we see the data in the tables;: a decrease in concentration and sperm count sperm count. Moreover when we look at the data in the overall and in the non responders: table 1 and 2

Overall concentration :           14.05 and then 10.0

Total count   31.66 and then 24.25

Non responders   concentration  14.05  and then 10.1 

Total count 31.66 and then 24.15

How is this possible if we have 15 responders+ and 18 non responders

E have a 45/55 ratio….it seems impossible to have the same values in the table

It looks for table 1 and 2 like a cut and paste

This opens the question, on which criteria the Responders++ and the responders—were selected. Especially when we look line 186-187 “ in the non responder group  (63.6 % ): this means 21 patients...

Hen several other points. For decondensation. The technique is not a standard, well recognize technique for sperm…Malondialdehyde is a good marker of lipid peroxidation, DNA fragmentation and decondensation.

The real interesting part is the polymorphism

In conclusion, this paper is not really informative, rather confusing and troubling. Totally impossible t understand table 1 and 2. The selection for Non responders and responders seems “moving” according  to the parameter tested

. Cannot be accepted without profound modifications

First of all the authors say” Scientific evidence……” , for LH and testosterone it is probably true but for FSH..

The least we can say it is rather uncertain:  for DFI decrease: Colacurci et al. :paper (abstract)     In the subgroup of patients with high basal DFI values (>15%), rFSH treatment significantly increased DFI (P < .01),This is not exactly what we can expect 

Santi study (abstract)…: limit the strength of these results.

This is obvious when we see the data in the tables;: a decrease in concentration and sperm count sperm count. Moreover when we look at the data in the overall and in the non responders: table 1 and 2

Overall concentration :           14.05 and then 10.0

Total count   31.66 and then 24.25

Non responders   concentration  14.05  and then 10.1 

Total count 31.66 and then 24.15

How is this possible if we have 15 responders+ and 18 non responders

E have a 45/55 ratio….it seems impossible to have the same values in the table

It looks for table 1 and 2 like a cut and paste

This opens the question, on which criteria the Responders++ and the responders—were selected. Especially when we look line 186-187 “ in the non responder group  (63.6 % ): this means 21 patients...

Hen several other points. For decondensation. The technique is not a standard, well recognize technique for sperm…Malondialdehyde is a good marker of lipid peroxidation, DNA fragmentation and decondensation.

The real interesting part is the polymorphism

In conclusion, this paper is not really informative, rather confusing and troubling. Totally impossible t understand table 1 and 2. The selection for Non responders and responders seems “moving” according  to the parameter tested

. Cannot be accepted without profound modifications

First of all the authors say” Scientific evidence……” , for LH and testosterone it is probably true but for FSH..

The least we can say it is rather uncertain:  for DFI decrease: Colacurci et al. :paper (abstract)     In the subgroup of patients with high basal DFI values (>15%), rFSH treatment significantly increased DFI (P < .01),This is not exactly what we can expect 

Santi study (abstract)…: limit the strength of these results.

This is obvious when we see the data in the tables;: a decrease in concentration and sperm count sperm count. Moreover when we look at the data in the overall and in the non responders: table 1 and 2

Overall concentration :           14.05 and then 10.0

Total count   31.66 and then 24.25

Non responders   concentration  14.05  and then 10.1 

Total count 31.66 and then 24.15

How is this possible if we have 15 responders+ and 18 non responders

E have a 45/55 ratio….it seems impossible to have the same values in the table

It looks for table 1 and 2 like a cut and paste

This opens the question, on which criteria the Responders++ and the responders—were selected. Especially when we look line 186-187 “ in the non responder group  (63.6 % ): this means 21 patients...

Hen several other points. For decondensation. The technique is not a standard, well recognize technique for sperm…Malondialdehyde is a good marker of lipid peroxidation, DNA fragmentation and decondensation.

The real interesting part is the polymorphism

In conclusion, this paper is not really informative, rather confusing and troubling. Totally impossible t understand table 1 and 2. The selection for Non responders and responders seems “moving” according  to the parameter tested

. Cannot be accepted without profound modifications

First of all the authors say” Scientific evidence……” , for LH and testosterone it is probably true but for FSH..

The least we can say it is rather uncertain:  for DFI decrease: Colacurci et al. :paper (abstract)     In the subgroup of patients with high basal DFI values (>15%), rFSH treatment significantly increased DFI (P < .01),This is not exactly what we can expect 

Santi study (abstract)…: limit the strength of these results.

This is obvious when we see the data in the tables;: a decrease in concentration and sperm count sperm count. Moreover when we look at the data in the overall and in the non responders: table 1 and 2

Overall concentration :           14.05 and then 10.0

Total count   31.66 and then 24.25

Non responders   concentration  14.05  and then 10.1 

Total count 31.66 and then 24.15

How is this possible if we have 15 responders+ and 18 non responders

E have a 45/55 ratio….it seems impossible to have the same values in the table

It looks for table 1 and 2 like a cut and paste

This opens the question, on which criteria the Responders++ and the responders—were selected. Especially when we look line 186-187 “ in the non responder group  (63.6 % ): this means 21 patients...

Hen several other points. For decondensation. The technique is not a standard, well recognize technique for sperm…Malondialdehyde is a good marker of lipid peroxidation, DNA fragmentation and decondensation.

The real interesting part is the polymorphism

In conclusion, this paper is not really informative, rather confusing and troubling. Totally impossible t understand table 1 and 2. The selection for Non responders and responders seems “moving” according  to the parameter tested

. Cannot be accepted without profound modifications

First of all the authors say” Scientific evidence……” , for LH and testosterone it is probably true but for FSH..

The least we can say it is rather uncertain:  for DFI decrease: Colacurci et al. :paper (abstract)     In the subgroup of patients with high basal DFI values (>15%), rFSH treatment significantly increased DFI (P < .01),This is not exactly what we can expect 

Santi study (abstract)…: limit the strength of these results.

This is obvious when we see the data in the tables;: a decrease in concentration and sperm count sperm count. Moreover when we look at the data in the overall and in the non responders: table 1 and 2

Overall concentration :           14.05 and then 10.0

Total count   31.66 and then 24.25

Non responders   concentration  14.05  and then 10.1 

Total count 31.66 and then 24.15

How is this possible if we have 15 responders+ and 18 non responders

E have a 45/55 ratio….it seems impossible to have the same values in the table

It looks for table 1 and 2 like a cut and paste

This opens the question, on which criteria the Responders++ and the responders—were selected. Especially when we look line 186-187 “ in the non responder group  (63.6 % ): this means 21 patients...

Hen several other points. For decondensation. The technique is not a standard, well recognize technique for sperm…Malondialdehyde is a good marker of lipid peroxidation, DNA fragmentation and decondensation.

The real interesting part is the polymorphism

In conclusion, this paper is not really informative, rather confusing and troubling. Totally impossible t understand table 1 and 2. The selection for Non responders and responders seems “moving” according  to the parameter tested

. Cannot be accepted without profound modifications

First of all the authors say” Scientific evidence……” , for LH and testosterone it is probably true but for FSH..

The least we can say it is rather uncertain:  for DFI decrease: Colacurci et al. :paper (abstract)     In the subgroup of patients with high basal DFI values (>15%), rFSH treatment significantly increased DFI (P < .01),This is not exactly what we can expect 

Santi study (abstract)…: limit the strength of these results.

This is obvious when we see the data in the tables;: a decrease in concentration and sperm count sperm count. Moreover when we look at the data in the overall and in the non responders: table 1 and 2

Overall concentration :           14.05 and then 10.0

Total count   31.66 and then 24.25

Non responders   concentration  14.05  and then 10.1 

Total count 31.66 and then 24.15

How is this possible if we have 15 responders+ and 18 non responders

E have a 45/55 ratio….it seems impossible to have the same values in the table

It looks for table 1 and 2 like a cut and paste

This opens the question, on which criteria the Responders++ and the responders—were selected. Especially when we look line 186-187 “ in the non responder group  (63.6 % ): this means 21 patients...

Hen several other points. For decondensation. The technique is not a standard, well recognize technique for sperm…Malondialdehyde is a good marker of lipid peroxidation, DNA fragmentation and decondensation.

The real interesting part is the polymorphism

In conclusion, this paper is not really informative, rather confusing and troubling. Totally impossible t understand table 1 and 2. The selection for Non responders and responders seems “moving” according  to the parameter tested

. Cannot be accepted without profound modifications

First of all the authors say” Scientific evidence……” , for LH and testosterone it is probably true but for FSH..

The least we can say it is rather uncertain:  for DFI decrease: Colacurci et al. :paper (abstract)     In the subgroup of patients with high basal DFI values (>15%), rFSH treatment significantly increased DFI (P < .01),This is not exactly what we can expect 

Santi study (abstract)…: limit the strength of these results.

This is obvious when we see the data in the tables;: a decrease in concentration and sperm count sperm count. Moreover when we look at the data in the overall and in the non responders: table 1 and 2

Overall concentration :           14.05 and then 10.0

Total count   31.66 and then 24.25

Non responders   concentration  14.05  and then 10.1 

Total count 31.66 and then 24.15

How is this possible if we have 15 responders+ and 18 non responders

E have a 45/55 ratio….it seems impossible to have the same values in the table

It looks for table 1 and 2 like a cut and paste

This opens the question, on which criteria the Responders++ and the responders—were selected. Especially when we look line 186-187 “ in the non responder group  (63.6 % ): this means 21 patients...

Hen several other points. For decondensation. The technique is not a standard, well recognize technique for sperm…Malondialdehyde is a good marker of lipid peroxidation, DNA fragmentation and decondensation.

The real interesting part is the polymorphism

In conclusion, this paper is not really informative, rather confusing and troubling. Totally impossible t understand table 1 and 2. The selection for Non responders and responders seems “moving” according  to the parameter tested

. Cannot be accepted without profound modifications

First of all the authors say” Scientific evidence……” , for LH and testosterone it is probably true but for FSH..

The least we can say it is rather uncertain:  for DFI decrease: Colacurci et al. :paper (abstract)     In the subgroup of patients with high basal DFI values (>15%), rFSH treatment significantly increased DFI (P < .01),This is not exactly what we can expect 

Santi study (abstract)…: limit the strength of these results.

This is obvious when we see the data in the tables;: a decrease in concentration and sperm count sperm count. Moreover when we look at the data in the overall and in the non responders: table 1 and 2

Overall concentration :           14.05 and then 10.0

Total count   31.66 and then 24.25

Non responders   concentration  14.05  and then 10.1 

Total count 31.66 and then 24.15

How is this possible if we have 15 responders+ and 18 non responders

E have a 45/55 ratio….it seems impossible to have the same values in the table

It looks for table 1 and 2 like a cut and paste

This opens the question, on which criteria the Responders++ and the responders—were selected. Especially when we look line 186-187 “ in the non responder group  (63.6 % ): this means 21 patients...

Hen several other points. For decondensation. The technique is not a standard, well recognize technique for sperm…Malondialdehyde is a good marker of lipid peroxidation, DNA fragmentation and decondensation.

The real interesting part is the polymorphism

In conclusion, this paper is not really informative, rather confusing and troubling. Totally impossible t understand table 1 and 2. The selection for Non responders and responders seems “moving” according  to the parameter tested

. Cannot be accepted without profound modifications

First of all the authors say” Scientific evidence……” , for LH and testosterone it is probably true but for FSH..

The least we can say it is rather uncertain:  for DFI decrease: Colacurci et al. :paper (abstract)     In the subgroup of patients with high basal DFI values (>15%), rFSH treatment significantly increased DFI (P < .01),This is not exactly what we can expect 

Santi study (abstract)…: limit the strength of these results.

This is obvious when we see the data in the tables;: a decrease in concentration and sperm count sperm count. Moreover when we look at the data in the overall and in the non responders: table 1 and 2

Overall concentration :           14.05 and then 10.0

Total count   31.66 and then 24.25

Non responders   concentration  14.05  and then 10.1 

Total count 31.66 and then 24.15

How is this possible if we have 15 responders+ and 18 non responders

E have a 45/55 ratio….it seems impossible to have the same values in the table

It looks for table 1 and 2 like a cut and paste

This opens the question, on which criteria the Responders++ and the responders—were selected. Especially when we look line 186-187 “ in the non responder group  (63.6 % ): this means 21 patients...

Hen several other points. For decondensation. The technique is not a standard, well recognize technique for sperm…Malondialdehyde is a good marker of lipid peroxidation, DNA fragmentation and decondensation.

The real interesting part is the polymorphism

In conclusion, this paper is not really informative, rather confusing and troubling. Totally impossible t understand table 1 and 2. The selection for Non responders and responders seems “moving” according  to the parameter tested

. Cannot be accepted without profound modifications